# Optical Monitoring of the Biodegradation of Porous and Solid Silicon Nanoparticles

**DOI:** 10.3390/nano11092167

**Published:** 2021-08-25

**Authors:** Maxim B. Gongalsky, Nikolay V. Pervushin, Daria E. Maksutova, Uliana A. Tsurikova, Pavel P. Putintsev, Oleg D. Gyuppenen, Yana V. Evstratova, Olga A. Shalygina, Gelina S. Kopeina, Andrey A. Kudryavtsev, Boris Zhivotovsky, Liubov A. Osminkina

**Affiliations:** 1Faculty of Physics, Lomonosov Moscow State University, 119991 Moscow, Russia; medar908@gmail.com (D.E.M.); natashina78@yandex.ru (U.A.T.); putintcev.pp19@physics.msu.ru (P.P.P.); guppenen19@mail.ru (O.D.G.); olga@ofme.phys.msu.ru (O.A.S.); 2Faculty of Medicine, Lomonosov Moscow State University, 119991 Moscow, Russia; rhododendron.nick@mail.ru (N.V.P.); lirroster@gmail.com (G.S.K.); boris.zhivotovsky@ki.se (B.Z.); 3Institute of Theoretical and Experimental Biophysics, Russian Academy of Sciences, 142290 Pushchino, Russia; yannaevstratova@gmail.com (Y.V.E.); centavr42@mail.ru (A.A.K.); 4Institute of Environmental Medicine, Karolinska Institutet, Box 210, 17177 Stockholm, Sweden; 5Institute for Biological Instrumentation, Russian Academy of Sciences, 142290 Pushchino, Russia

**Keywords:** silicon nanoparticles, porous nanoparticles, solid nanoparticles, biodegradation, Raman, photoluminescence, cytotoxicity, biophotonics

## Abstract

Silicon nanoparticles (SiNP) are currently of great interest, especially in biomedicine, because of their unique physicochemical properties combined with biodegradability. SiNPs can be obtained in various ways and can have either a non-porous solid (sol-) or porous (por-) structure. In this work, we carry out detailed optical monitoring of sol- and por-SiNP biodegradation using Raman and photoluminescence (PL) micro-spectroscopy. SiNPs were obtained by ultrasound grinding of sol- or por-silicon nanowires, created by silver-assisted chemical etching of crystalline Si with different doping levels. In this case, sol-SiNPs consist of nanocrystals 30 nm in size, while por-SiNPs consist of small 3 nm nanocrystals and 16 nm pores. Both SiNPs show low in vitro cytotoxicity towards MCF-7 and HEK293T cells up to 800 μg/mL. The appearance of the F-band (blue–yellow) PL, as well as a decrease in the intensity of the Raman signal, indicate the gradual dissolution of the sol-SiNPs during 20 days of incubation. At the same time, the rapid dissolution of por-SiNP within 24 h is identified by the quenching of their S-band (red) PL and the disappearance of the Raman signal. The obtained results are important for development of intelligent biodegradable drug delivery systems based on SiNPs.

## 1. Introduction

Silicon nanoparticles (SiNPs) are attracting the scientific attention due to their various emerging biomedical applications [1,2]. Both porous (por-SiNPs) and solid (sol-SiNPs) nanoparticles are interesting, however their properties can vary significantly. SiNPs are promising theranostics agents [1] because they can combine both therapeutical and diagnostic modalities. Potential therapeutical effects include drug delivery [3], photodynamic therapy [4,5], enhancement of ultrasonic [6] and radiofrequency irradiations [7,8], etc. Diagnostics is provided by SiNPs themselves as contrast agents for such methods as fluorescent diffuse tomography [3], magnetic resonance imaging [9], and ultrasonic imaging [10]. The unique features of SiNPs are their biocompatibility [11] and biodegradability [3,12]. First is based on non-toxic nature of silicon itself since it takes significant part in human metabolism. Biodegradability is based on dissolution of SiNPs in conditions corresponding to living systems. Though dissolution is rather slow, the enormously large specific surface area of SiNPs (especially por-SiNPs) facilitates typical lifetimes in the range from several hours to several days [13,14]. The product of the dissolution is silicic acid (SiOH_4_), which is non-toxic in relevant concentrations and is easily excreted from organism.

However, improper usage of SiNPs may be dangerous, i.e., SiNPs may potentially agglomerate, facilitate blood clots formation, interfere with cell mitosis; excessive formation of silicic acid may significantly change pH value [6,11]. Therefore, precise monitoring of SiNPs biodegradation is required for successful in vivo and future clinic applications.

The gold standard method is inductively coupled plasma atomic emission spectroscopy (ICP-AES), which gives precise amount of Si in supernatant of centrifugated SiNPs suspension [3,13]. These data represent amount of the formed silicic acid and perfectly describes dissolution of SiNPs. However, the method is applied only for basic model systems, and it is useless for monitoring of SiNPs biodegradation in vitro and in vivo. For this purposes a couple of optical methods were proposed, i.e., Raman and photoluminescent (PL) spectroscopy [15,16]. Both methods are noninvasive and suitable for in vitro as well as in vivo investigation, although they provide indirect information of the remaining Si content, which requires accurate interpretation.

The optical methods allow to estimate both Si amount and average size of Si nanocrystals (nc-Si), which forms por-SiNPs. In general, size of nc-Si can be determined by position of both Raman and PL specific bands [17,18]. Moreover, Raman spectra show amount of amorphous Si (a-Si), which is known to be intermediate state of degrading SiNPs and indirect evidence of the biodegradation process [15].

SiNPs can be obtained from big variety of silicon-based materials and demonstrate different properties. However, all of them can be considered as por-SiNPs or sol-SiNPs, therefore that two types of nanomaterials have been chosen for present articles. Here, we utilized SiNPs obtained from Si nanowires created by metal-assisted chemical etching of crystalline Si wafers with different doping levels [19].

The present article shows how dual, Raman and luminescent, noninvasive optical monitoring of SiNPs biodegradation via their physicochemical characterizations on the different stages of that process can be made for two most important cases: porous or solid silicon-based nanoparticles.

## 2. Materials and Methods

### 2.1. Fabrication of Porous and Solid Silicon Nanoparticles

Both sol-SiNPs and por-SiNPs were obtained by procedure schematically shown in Figure 1. The only significant difference was specific resistivity (doping level) of the initial crystalline Si (c-Si) wafers. The formation procedure includes metal-assisted chemical etching (MACE) of c-Si to obtain silicon nanowires (SiNWs) and subsequent ultrasonic grinding of the SiNWs to obtain SiNPs.

As the first step of MACE, a (100)-oriented single crystalline silicon wafer was immersed in a mixture of 0.01 M AgNO_3_ and 5 M of HF in a volume ratio of 1:1 for 15 s. As a result, Ag nanoparticles (AgNPs) were sedimented on the surface of the c-Si wafer (shown as gray circles). Then, the c-Si wafer was placed into a mixture of 5 M HF and 30% H_2_O_2_ in the volume ratio 10:1 for 1 h. The layer of SiNWs occurred because of Ag nanoparticles-assisted chemical etching. Then, the samples were rinsed 3 times in bidistilled water and dried at room temperature. Etching of highly-doped (HD) c-Si with resistivity of 0.001 Ohm*cm resulted in formation of porous nanowires, while etching of low doped (LD) c-Si with resistivity of 1 Ohm*cm leads to formation of solid (non-porous) nanowires. That property of SiNWs is inherited by the resulting SiNPs.

Then, after the SiNW layer is formed, AgNPs are removed by dissolution in HNO_3_. Then, Si wafers with the SiNWs is subjected to ultrasonic treatment for 12 h, which results in detaching of the nanowires from the substrate and their fracturing into nanoparticles with a low aspect ratio. Finally, after several centrifugation steps, SiNPs aqueous suspension is acquired.

### 2.2. Characterization of Nanoparticles

Structural analysis of the samples was carried out by Carl Zeiss ULTRA 55 scanning electron microscopy (SEM) for SiNW layers and by LEO912 AB OMEGA transmission electron microscopy (TEM) for SiNPs. The samples of NPs for TEM studies were prepared by deposition of the nanoparticle powder on the standard carbon-coated gold TEM-grids. TEM images were processed with ImageJ Software to obtain the silicon nanocrystals (nc-Si) size distribution. To determine the hydrodynamic size of the nanoparticles a Malvern Instruments Mastersizer 200 have been used. The specific surface area of the SiNPs as well as their pore sizes and pore volumes were determined with N_2_ adsorption/desorption (Micromeritics Tristar 3000). The pore area from the adsorption branch using BET theory (Brunauer, Emmett, and Teller), and the pore size was calculated from the desorption branch using BJH theory (Barrett, Joyner, and Halenda). Before measurements, the samples were dried by lyophilization. Fourier transform infrared (FTIR) spectra were measured by using Bruker spectrometer and attenuated total reflectance (ATR) addon.

### 2.3. Dissolution in Model Liquids

SiNPs were introduced into phosphate-buffered saline (PBS, pH 7.2) to achieve the concentration of 0.5 mg/mL, then 5 mL of nanoparticles suspension were placed in dialysis bag (4.5 kDa pore diameter) and then in 200 mL of PBS at 37 °C (Binder chamber).

To measure the photoluminescence (PL) or Raman spectra, 0.1 mL of the suspension was dried on a metal plate after different time intervals of the incubation. The PL spectra of the samples were measured under excitation with a cw Ar-ion laser at 364 nm (power of 10 mW, spot diameter of 1 mm). The PL signal was detected using a grating monochromator (MS750, SOLAR TII, Minsk, Belarus) equipped with a CCD array. Raman spectra were measured using a Confotec™ MR350 confocal Raman microscope with laser excitation at 633 nm and a weak power of 1 mW to protect samples from overheating. MagicPlot software was used for deconvolution of spectra.

### 2.4. Cytotoxicity Studies

#### 2.4.1. Cell Culture and Experimental Procedures

The studies were performed using breast carcinoma MCF-7 cells and human embryonic kidney HEK293T cells, which were kindly provided by the Departament of Toxicology Karolinska Institutet (Stockholm, Sweden). All cells were grown in 5% CO_2_ at 37 °C, in DMEM (Gibco, Waltham, MA, USA) medium containing 4.5 g/L glucose (Gibco), 10% FBS (Gibco), in the presence of a mixture of antibiotics and antimycotics (Gibco). All cells were maintained in the logarithmic growth phase for experiments. After 24 h, the culture medium was replaced with fresh one for treatments. Before the treatment with nanoparticles, staurosporine or MG132 (both–Sigma, St. Louis, MO, USA) the culture medium was removed, and cells were washed with PBS. Fresh serum containing medium (DMEM supplemented with 10% FBS, 1% mixture of antibiotics and antimycotics) was added to cells. All cells were treated with nanoparticles (por-SiNPs and sol-SiNPs) in concentrations of 400 and 800 µg/mL and incubated for 24 h. Additionally, MCF-7 cells were treated with staurosporine in concentration 0.25 µM (4 h) and HEK293T cells were treated with MG132 in concentration 2 µM (24 h). Both agents were used as inductors of apoptosis.

#### 2.4.2. FACS Analysis

After the indicated time of treatment, cells were detached from the dishes using 0.25% trypsin (Gibco) and transferred into a conditioned medium. Then, cells were centrifuged (200–300 g, 5 min, 4 °C), washed twice with cold PBS solution (Paneco, Singapore), and the pellet was resuspended in PBS (100 μL per 1 million cells). Approximately 10^5^ cells (10 μL) were transferred into 200 μL of 1× annexin-binding buffer (BD Biosciences, San Jose, CA, USA) and 2 μL of Annexin V-FITC (Invitrogen) was added. Then, samples were incubated in the dark at room temperature for 15 min. Next, 5 μL of propidium iodide (50 μg/mL) (BD Biosciences) was added to each sample. The analysis of the cell population after 5 min of incubation in the dark at room temperature was performed using a BD FACS Canto II flow cytometer (BD Biosciences).

#### 2.4.3. Gel Electrophoresis and Western Blot Analysis (WB)

After the indicated time of treatment, cells were detached with scraper in the conditioned medium. The cells were centrifuged (300 rcf, 5 min, 4 °C) and washed twice with ice-cold PBS (Paneco). Then, the cell pellet was lysed in RIPA buffer (25 mM Tris-HCl (pH 7.4), 150 mM NaCl, 0.1% SDS, 0.5% sodium deoxycholate, 1% NP-40, cOmplete™ Protease Inhibitor Cocktail (Roche, Basel, Switzerland)), and incubated on ice for 20 min. After centrifugation (16,000 *g*, 15 min, 4 °C), a part of the supernatant was taken for protein concentration assay using the Pierce BCA Protein Assay Kit (Thermo Scientific, Waltham, MA, USA). Then, samples were mixed with Laemmli’s loading buffer and boiled for 7 min. Equal amount (20–40 μg) of protein extracts was separated using sodium dodecyl sulfate polyacrylamide gel electrophoresis (SDS-PAGE, 12% gel) at 100 mV, followed by blotting onto nitrocellulose membranes for 2 h at 110 V. After transferring, membranes were blocked for 1 h with 5% non-fat milk in tris-buffered saline (TBS) at room temperature and subsequently probed overnight with appropriate primary antibodies (1:1000), followed by incubation with a horseradish peroxidase-conjugated secondary antibody (from 1:2500 to 1:5000) for 1 h with 2.5% non-fat milk in TBS at RT. Detection was visualized with ECL (Amersham Biosciences, Piscataway, NJ, USA) and a ChemiDoc MP Imaging System (BioRad, Hercules, CA, USA).

#### 2.4.4. Antibodies

The following primary antibodies were used: poly (ADP-ribose)-polymerase (PARP, Abcam, ab137653, Cambridge, UK), tubulin-alpha (Abcam, ab7291), anti-rabbit full and cleaved caspase-3 (Cell Signaling, Danvers, MA, USA, 9662 s). Anti-rabbit or anti-mouse IgG, conjugated with horseradish peroxidase (Abcam, ab97200, and ab97046, respectively) were used as secondary antibodies.

### 2.5. SiNPs Biodegradation Studies

#### 2.5.1. Luminescent Imaging

Confocal luminescent imaging experiments were performed by using a TCS SP5 Leica confocal laser scanning microscope with excitation at 405 nm. The SiNP PL signal was recorded in the range of 510 to 800 nm. An oil immersion Leica 40×/1.25–0.75 objective was used for imaging.

Cells were grown on coverslips in a standard 10 cm Petri dishes filled with the DMEM culture medium. When cells reached the desired confluency, the media was removed from the dish and cells were washed with PBS. Fresh serum containing medium (DMEM supplemented with 10% FBS, 1% mixture of antibiotics and antimycotics) was added to cells. All cells were treated with nanoparticles (por-SiNPs and sol-SiNPs) in concentration 800 µg/mL and incubated for indicated time: por-SiNPs—9 and 24 h; sol-SiNPs—24 h and 7 days. After selected time of incubation, all cells were washed two times for 3 min with PBS and were fixed in 4% paraformaldehyde solution in PBS during 7 h at +4 °C.

#### 2.5.2. Raman Imaging and Spectral Data Analysis

Micro-Raman spectroscopy data were acquired using a confocal Raman microscope Confotec™ MR350, with laser excitation at 633 nm wavelength and 0.5 mW power. The MCF-7 and HEK293T cells were fixed on special CaF_2_ slides aiming to prevent a background scattering. Cells were grown in DMEM culture medium. When cells have reached the desired confluency, the media was removed from the dish and cells were washed with PBS. Fresh serum containing medium (DMEM supplemented with 10% FBS, 1% mixture of antibiotics and antimycotics) was added to cells. All cells were treated with nanoparticles (por-SiNPs and sol-SiNPs) in concentration 800 µg/mL and incubated for indicated time: por-SiNPs—6 and 24 h; sol-SiNPs—24 h and 7 days. After a selected time of incubation, all cells were washed two times for 3 min with PBS and were fixed in 4% paraformaldehyde solution in PBS during 7 h at +4 °C. Afterwards, the glasses with the cells were placed in standard 5 cm Petri dishes filled with distilled water. This prevents the cells from overheating during the measurements due to high laser power and enables longer imaging of the cell clusters. For this purpose, a 63×/NA 1.0 water dipping objective Zeiss was used to focus the laser on the cell. The obtained images of cell and intracellular uptake of nanoparticles were generated by applying a CCD camera Andor operating at −60 °C.

## 3. Results and Discussion

### 3.1. Sample Characterization

Porous and non-porous SiNWs were produced by silver-assisted chemical etching of high-doped (HD) and low-doped (LD) c-Si substrate, correspondingly. The structural difference between two types of nanowires can be explained by the doping level of the c-Si substrates used [19]. HD p-type c-Si contains of ~10^20^ free holes, which are necessary for the formation of pores inside the nanowires during the etching process. Therefore, porosification affects the whole volume of HD nanowire, in contrast to LD nanowire (LD p-type c-Si contains of ~10^15^ free holes), where only surface is partially etched (Appendix A, Appendix A). After ultrasonic grinding of nanowires, the resulting nanoparticles retain their structural properties and are porous or non-porous.

The obtained TEM images of sol-SiNPs and por-SiNPs are shown in Figure 2A,D correspondingly. Diffraction patterns are shown in the insets. The solid (denser) origin of sol-SiNPs is clearly visible, as well as the porous structure of por-SiNPs.

TEM images allow estimating the typical size of SiNPs, which is in the range of 100 to 300 nm and corresponds to the DLS measurements (Figure 2C,F). Diffraction patters show significant difference between sol-SiNPs and por-SiNPs. Sol-SiNPs exhibit pronounced isolated narrow diffraction peaks corresponding to small amount of randomly oriented relatively big (>7 nm) nc-Si, which shows solid character of sol-SiNPs. By contrast, por-SiNPs exhibit several broadened concentrated rings corresponding to huge amount of small (<5 nm) nc-Si. Broadening of peaks is explained by Debye-Sherrer effect in small crystallites [20]. That also points to porous structure of por-SiNPs as the size of SiNPs is much bigger than size of nc-Si.

Dark-field TEM image of SiNPs tuned to a certain diffraction peak (Appendix A, Appendix A) was used for calculation of nc-Si size distribution in SiNP samples (Figure 2B,E). The average nc-Si diameter was found to be 30 nm and 3 nm for sol-SiNPs and por-SiNPs, correspondingly. These quantitative results are in accordance with conclusions made above from qualitative diffraction patterns.

Additionally, nitrogen gas adsorption/desorption analysis was used to determine the specific surface area, pore volume, and pore diameters of the obtained samples (Table 1). Large values of the specific surface area and pore volume of por-SiNPs confirm the porous structure of these samples, the average pore diameter in which is 16 nm. At the same time, a small surface area indicates a non-porous structure of sol-SiNPs, and the resulting pore diameter and volume correspond, apparently, to the agglomeration of nanoparticles upon drying.

FTIR spectra of both sol-SiNPs and por-SiNPs are shown in Appendix A. Spectra exhibit prominent wide Si-O-Si band at 1100 cm^−1^, which is attributed to thick oxide layer at the surface of SiNPs [21]. Mostly, oxide appears due to HNO_3_ treatment aimed to dissolve Ag nanoparticles inside Si wafers after MACE-process. Therefore, both samples are well oxidized, which provides them good hydrophilic properties and similar surface composition.

### 3.2. Optical Monitoring of SiNPs Dissolution in Model Liquids

SiNPs exhibit unique optical properties, which were used for monitoring of their dissolution process. First of all, the modeling experiments of SiNPs incubation in dialysis bag in PBS at 37 °C were carried out (Figure 3A). The experiments emulate intracellular conditions, i.e., pH, ionic strength, and temperature of the medium, and dialysis bag emulates cell metabolism: it is implied that formed silicic acid is quickly removed from vicinity of SiNPs by cell self-regulation. The size of pores in dialysis bag (4.5 kDa) was chosen to allow easy penetration of silicic acid molecules (green circles in Figure 3A) and PBS ions (gray circles) but deny penetration of SiNPs (red circles).

Figure 3B shows modification of PL and Raman spectra of the samples during the experiment. In general, PL of SiNPs can be caused by (1) quantum confinement of excitons in small nc-Si (<5 nm) and (2) defect states in a-Si or amorphous silicon dioxide/suboxide. The first type of PL exhibits correlation between nc-Si size and PL peak position, unlike the second type, which depends mostly on chemical composition, stoichiometry, cross-phase interfaces, etc. Spectral location is also different, i.e., excitonic S-band PL is registered in red 1.3–2.2. eV range [17,18], while defect F-band PL generally is in blue–yellow 1.8–3.5 eV range, though some overlapping may occur. Thus, PL of por-SiNPs as well as initial sol-SiNPs can be considered as excitonic. This band quickly disappears after incubation of sol-SiNPs, because they are non-homogeneous, i.e., thin layer of small nc-Si suitable for excitonic PL is present on the surface of the nanoparticles, but it quickly dissolves, and the band vanishes.

The high-energy band (2.3 eV), which is slightly pronounced in PL spectrum of initial sol-SiNPs, dramatically increases with incubation time. It is attributed to an amorphous intermediary layer, which can be considered as a shell of big non-fluorescent nc-Si. Changes of PL intensity show that this layer is negligible in initial sol-SiNPs, then grows during the incubation (~20 days), and finally starts to dissolve after the whole Si core is converted into amorphous phase (after 20 days).

Similar processes take place with por-SiNPs, i.e., reduction of excitonic band and enhancement of defect band. However, por-SiNPs structure is different: they are consisted of much bigger amount of small nc-Si with quantum confinement and there is no big nc-Si suitable for growing of thick amorphous layer. Therefore, the low-energy band (1.7–1.8 eV) vanishes along with dissolution of small nc-Si, but some of them nevertheless convert into amorphous nanoparticles and contribute to high energy bands (2.5–3.5 eV). Note that full decomposition of typical PL spectra of por-SiNPs is shown in Appendix A. Typical external quantum yield of PL is about 2% and 0.2% for as-prepared por-SiNPs and sol-SiNPs, respectively [22].

All excitonic bands of PL can be used for estimation of average nc-Si diameter in the samples. According to quantum confinement models [17] peak positions for sol-SiNPs and por-SiNPs give 5 nm and 3 nm, respectively. However, this evaluation for sol-SiNPs cannot be applied for estimation of nc-Si in the whole sample, because it characterizes only tiny fraction of small nc-Si presented on surface of big SiNPs, while the majority of Si structures does not emit any PL. On contrary, estimations made for por-SiNPs give relevant value in a good accordance with TEM dark field results. The degradation of SiNPs is accompanied with shrinking of nc-Si core size (from 3 nm to 2.5 nm), which is visible by slight shift of excitonic PL band towards higher energies.

Raman spectra also allow monitoring of SiNPs biodegradation. Crystalline silicon exhibits narrow Raman band at 520.5 cm^−1^. Intensity of this line is proportionate to the volume of scattering Si. Nanocrystalline Si (with nc-Si diameter < 7 nm) is characterized by a similar line but broadened and shifted depending on nc-Si sizes and dispersion of nc-Si ensemble [17]. According to that, nc-Si in sol-SiNPs are too big to be described within that approach, and the Raman band does not exhibit any significant confinement-based shift (Figure 3C). These data are consistent with both TEM and PL measurements for sol-SiNPs. Note, that despite during biodegradation of sol-SiNPs some amorphous layer and small nanocrystals may occur, their contribution into the overall Raman signal is negligible. Therefore, Raman band does not change its position and could not be used for control of the biodegradation process. Thus, Raman intensity was used instead of Raman shift (Figure 3B).

In the Raman spectra of por-Si nanoparticles (Figure 3E), after 1 and 6 h of incubation, a low-frequency shift of the maximum is observed in comparison with the c-Si band, the spectrum broadens, and a shoulder appears at 480 cm^−1^ corresponding to light scattering in amorphous silicon. The diameter of nc-Si in por-SiNPs, calculated from the position of the Raman spectra maximum [15] was about 4 nm after 1 h of incubation, and about 3 nm after 6 h. The absence of the Raman signal after 24 h of incubation indicates complete dissolution of por-SiNPs.

Therefore, both Raman and PL spectra show that typical dissolution time is about 15–21 days for sol-SiNPs and 24 h for por-Si. The difference is explained by different nc-Si sizes, and also their different specific surface area which is an important factor dissolution rate [14].

### 3.3. Cytotoxicity Study

To study whether nanoparticles (por-SiNPs and sol-SiNPs) in concentration 400 and 800 µg/mL cause death of human embryonic kidney HEK293T cells (shown below) and breast carcinoma MCF-7 cells (shown in Appendix A), Western blot analysis was used. Cells were incubated with the nanoparticles for 1, 2, and 3 days. The effector caspase-3 autocatalytic cleavage and activation and PARP degradation is well-known apoptotic markers [23]. Note that cleavage of PARP, which is involved in the process of DNA damage repair, by effector caspase-3 leads to its inactivation and indicates the intensity of apoptosis in cells [23]. Moreover, caspase-3 activity is essential for the majority of morphological and biochemical apoptosis-associated events [24]. According to WB analysis, these nanoparticles did not show apoptotic response in HEK293T cells: por-SiNPs and sol-SiNPs did not led to a generation of caspase-3 p17/19 active fragment and subsequent cleavage of its substrate–PARP (Figure 4). Similar effects were observed in caspase-3 deficient MCF-7 cells [25]; neither NP caused cleavage of PARP (Appendix A) [26]. These results were supported by data obtained using flow cytometric analysis with double-staining using Annexin V–FITC in combination with propidium iodide (PI) [24]. This approach is commonly applied to evaluate the population of apoptotic and necrotic cells. Early apoptotic cells are stained with annexin V-FITC only (Annexin V/PI +/−), late apoptotic cells are stained with Annexin V/PI+/+, necrotic cells were stained with PI only (Annexin V/PI −/+) and viable cells are negative to this staining. Cell death analysis using Annexin V-FITC/PI staining have revealed that por-SiNPs and sol-SiNPs after 1, 2, and 3 days of incubation in selected concentrations did not decrease cell viability compared to control cells (Figure 4).

A broad-spectrum protein kinase inhibitor, Staurosporine (STS), in concentration 0.25 μM for 4 h and a proteasome inhibitor MG132 in concentration 2 μM for 24 h were used as positive controls for apoptosis induction in MCF-7 and HEK293T cells, respectively. MG132 induced activation of effector caspase-3 and PARP cleavage in HEK293T cells, and STS caused cleavage of PARP in MCF-7 cells. Both agents decreased cell viability to 77% and 74% compared to non-treated cells in MCF-7 and HEK293T cells, correspondingly (Figure 4 and Appendix A).

### 3.4. Optical Monitoring of SiNPs Biodegradation In Vitro

Figure 5 shows widefield images and Raman spectra of SiNPs inside MCF7 cells (similar data for HEK293T cells is shown in Appendix A). Images of sol-SiNPs were obtained after 1 and 7 days of incubation with cells. One day was sufficient for significant internalization of sol-SiNPs inside cells, while their biodegradation was suggested to be negligible. Indeed, there are many dark spots corresponding to sol-SiNPs on the image, which is confirmed by strong Raman absorption band at 520.5 cm^−1^. The next image shows that after 7 days of incubation, the concentration of sol-SiNPs is much smaller and the Raman signal is weaker. However, it is still clearly pronounced, and no shift of the band is detected; this is consistent with previous results obtained for PBS. Note, that internalization of SiNPs inside the cells was verified by simultaneous presence of both nc-Si band and cell components bands (mostly by proteins and lipids) in the Raman spectra (Appendix A).

Por-SiNPs have less contrast on widefield images due to high porosity, quantum confinement, and subsequently low absorption coefficient [21]. Thus, only some black spots are visible on the images, and not all of them are por-SiNPs as it was justified by their Raman spectra. The reason for that is the much smaller scattering cross section due to both lower absorption and decrease of the oscillator strength caused by phonon confinement [27]. The typical spectrum of por-SiNPs is a superposition of broadened and shifted nc-Si band and less pronounced a-Si band located around 480 cm^−1^, which points to partial biodegradation of por-SiNPs. Minimal time, required for efficient internalization of por-SiNPs was 6 h, therefore imaging of initial por-SiNPs was not possible.

Complete biodegradation of por-SiNPs after 24 h of incubation was observed. All dark spots were checked and no evidence for nc-Si Raman band was found. However, some very small por-SiNPs (as small as single nc-Si) can be present, according to PL spectra (see Figure 3). Such small and strongly degraded SiNPs have no contrast in Raman imaging, while residual PL emission can be observed. The data obtained by Raman imaging of por-SiNPs in cells are also consistent with experiment for PBS.

Luminescent imaging also allows to visualize SiNPs in cells. However, the method is better applicable to por-SiNPs due to the larger amount of tiny luminescent nc-Si in them than for sol-SiNP. Figure 6 shows widefield view of the HEK293T cells (first column), fluorescence (second column), merged images (third column), and the selected magnification of the merged images (fourth column).

Sol-SiNPs internalized after 1 day exhibit weak luminescence, therefore, they are visible only by adsorption on widefield images. However, after 7 days of incubation with cells, PL increases and sol-SiNPs become more visible in fluorescence mode.

On contrary, por-SiNPs are in a great abundance inside cells cytoplasm just after 9 h of incubation. After 24 h, almost complete biodegradation of por-SiNPs occurs and their PL signal is much weaker, as it is seen in Figure 6. Similar results were obtained with MCF-7 cells (Appendix A).

The important question for correct interpretation of the obtained results is verification of successful internalization of SiNPs inside cells. As it may be difficult to see internalization directly in presented Figure 5 and Figure 6, additional images are shown in Appendix A. Appendix A demonstrates three cross-sectional views of a 3D scan, which shows that SiNPs are located inside the cells, not at the surface of Petri dish or above cells. Most often, SiNPs are localized in the area surrounding the cell nuclei in the vicinity of endoplasmic reticulum. Cell are visualized by their autofluorescence. Another proof of SiNPs internalization is based on integrated luminescence depth profile. Appendix A shows that maximal PL intensity from SiNPs is reached approximately at the depth, corresponding to the average center of cells. If SiNPs are deposited on the Petri dish bottom or on top of cells, maximum of PL will be at highest or lowest «z», but not in between.

## 4. Conclusions

Thus, two optical noninvasive methods of monitoring of SiNPs biodegradation were developed. They were applied to two types of nanoparticles, i.e., solid and porous nanoparticles, which can be considered as reference materials for all variety of silicon-based nanomaterials. Both samples were prepared by similar etching procedures of low- and high-doped Si wafers in order to minimize influence of other factors. Ultrasonic grinding of the obtained Si nanowires formed 140 nm sol-SiNPs and 130 nm por-SiNPs consisted of 30 nm and 3 nm Si nanocrystals, respectively.

Structural differences lead to optical ones. Por-SiNPs were found to be contrast in luminescent bioimaging, while sol-SiNPs have better contrast in Raman spectroscopy. This was explained by quantum confinement effects and high porosity of por-SiNPs, which resulted in efficient S-band PL in red range, originated from excitons confined in small nanocrystals. A similar band was found for sol-SiNPs, but it vanished after 1–2 h of incubation and attributed to minority of small nanocrystals on the surface of the nanoparticles. Further luminescent properties of sol-SiNPs are attributed to F-band (blue–yellow range) PL originated from defects in emerging oxide shell during their biodegradation. The optical study showed that typical biodegradation lifetimes were 20 days for sol-SiNPs and only 24 h for por-SiNPs. This difference is caused by different nc-Si sizes, which is proportionate to the biodegradation rate. The study included SiNPs dissolution in dialysis bag in phosphate buffer saline at 37 °C (emulation of biological fluid) and spectral-resolved Raman measurements and photoluminescent bioimaging of SiNPs inside HEK293T and MCF-7 cells. Both experiments showed similar results.

Both types of SiNPs exhibited no toxicity towards MCF-7 and HEK293T cells in concentration up to 800 μg/mL. Western blot analysis revealed that both por-SiNPs and sol-SiNPs did not led to a generation of caspase-3 p17/19 active fragment and subsequent cleavage of its substrate–PARP, which pointed to the absence of necrosis and apoptosis. These results were supported by flow cytometric analysis with double-staining method (Annexin V–FITC and propidium iodide (PI)), which also showed negligible population of apoptotic and necrotic cells after 1, 2, and 3 days of incubation.

Thus, it is possible to tailoring the rate of biodegradation of nanoparticles by changing the size of their nanocrystals. Two complimentary optical methods can be used for biodegradation monitoring of the variety of silicon-based materials. Raman imaging is most applicable for those with big (>5 nm) nanocrystals, while photoluminescent imaging is suitable for materials with small (<5 nm) nanocrystals. The results obtained can be useful for the development of intelligent drug delivery systems based on silicon nanoparticles with a tailored biodegradation time for sustained drug release and the development of new approaches in theranostics of diseases.

## Figures and Tables

**Figure 1 nanomaterials-11-02167-f001:**
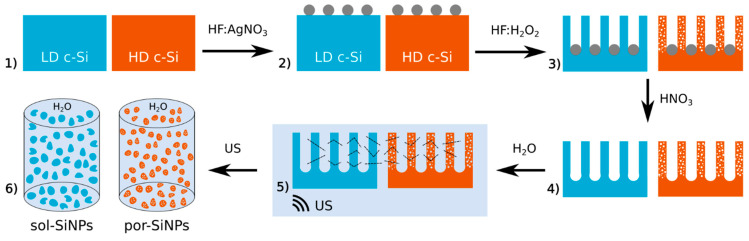
Schematic representation of both sol-SiNPs (blue) and por-SiNPs (brown) formation. Stages followed by arrows: (1) bulk low-doped (LD) and high-doped (HD) c-Si wafers; (2) Ag nanoparticles (gray circles) deposited on the c-Si surfaces; (3) SiNW layers formed by Ag-assisted chemical etching of the c-Si wafer with remnant AgNPs at the bottom of the layer; HD SiNWs exhibit porous structure; (4) SiNW layers after dissolution of AgNPs; (5) SiNWs under ultrasonic (US) treatment, dashed lines show fractures caused by US; (6) The resulting aqueous suspensions of sol- and por-SiNPs. Arrows are accompanied by chemical formula of reagents/medium.

**Figure 2 nanomaterials-11-02167-f002:**
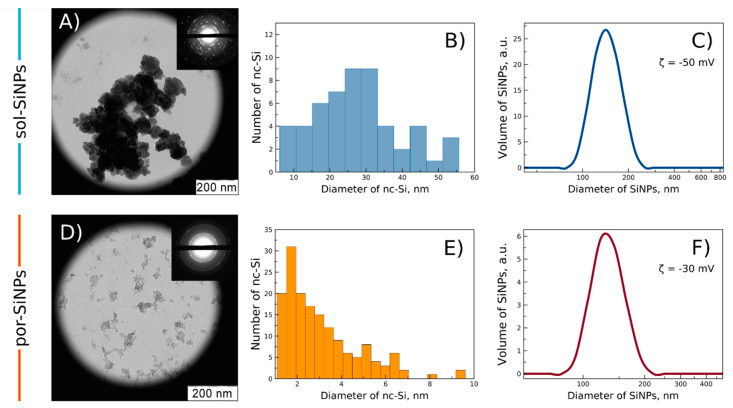
Characterization of SiNPs. TEM images of (**A**) sol-SiNPs and (**D**) por-SiNPs. The insets show corresponding diffraction patterns. Nanocrystal size (nc-Si) distribution of (**B**) sol-SiNPs and (**E**) por-SiNPs, obtained from dark-field TEM images. Dynamic light scattering data on SiNPs: hydrodynamic diameter distributions for sol-SiNPs (**C**) and por-SiNPs (**F**).

**Figure 3 nanomaterials-11-02167-f003:**
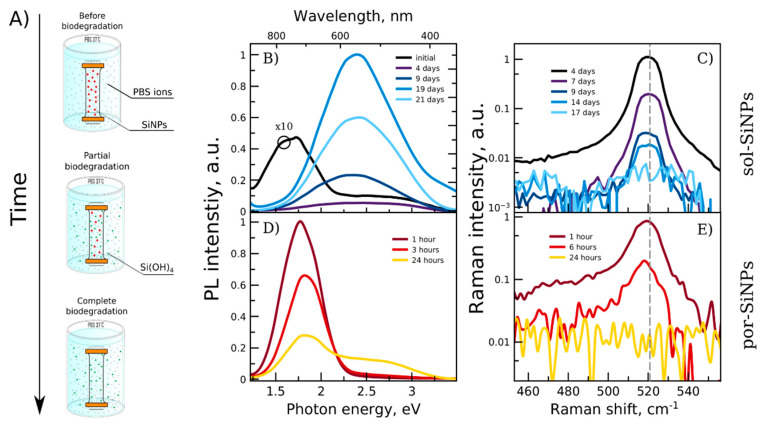
(**A**) Schematic view of dissolution of SiNPs in a dialysis bag from initial state (top) to completely dissolved (bottom). Red circles—SiNPs; green circles—silicic acid molecules, Si(OH)_4_; gray circles—PBS ions. Photoluminescence (**B**,**D**) and Raman (**C**,**E**) spectra of sol-SiNPs (**B**,**C**) and por-SiNPs (**D**,**E**) during degradation in PBS (37 °C). Legends show incubation time. Black PL curve for initial sol-SiNPs (plot B) was multiplied by a factor of 10. Vertical dashed gray line points to 520.5 cm^−1^ corresponding to position of c-Si Raman band.

**Figure 4 nanomaterials-11-02167-f004:**
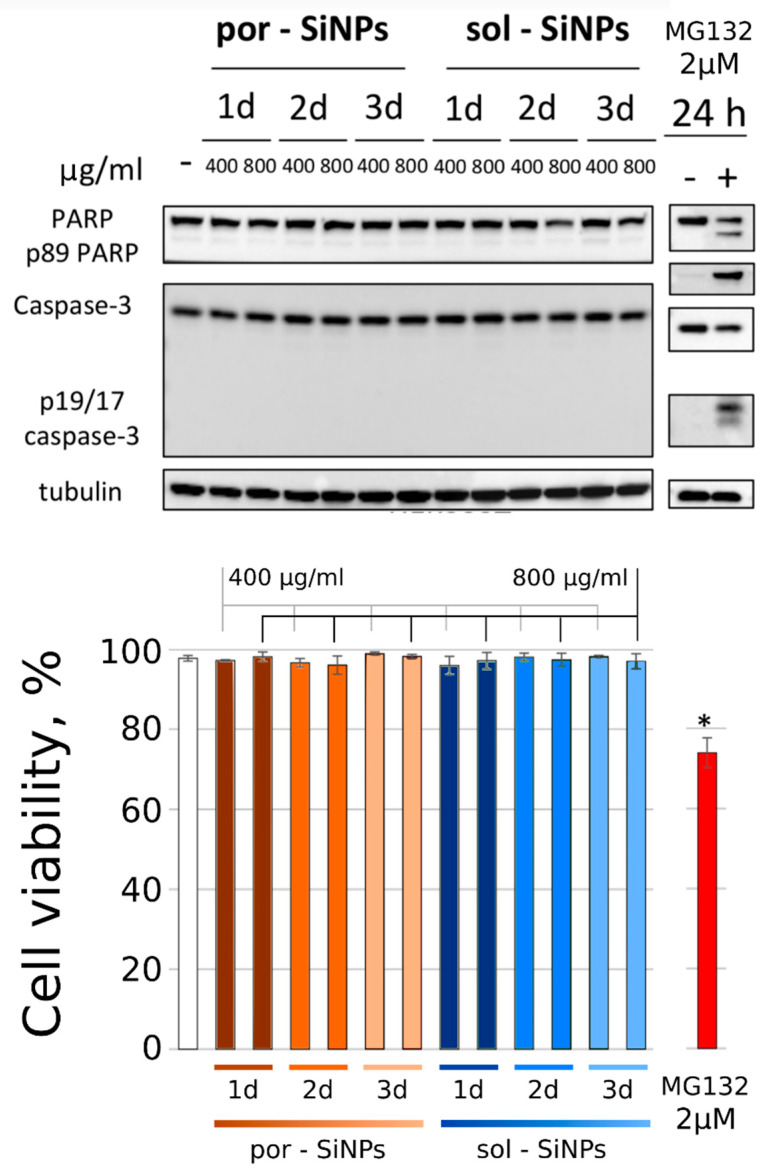
Western blot analysis of total cellular lysates from HEK293T cells upon treatment with two types of nanoparticles (por-SiNPs and sol-SiNPs) and positive control for apoptosis induction–MG132. Tubulin was used as a loading control. Designations: PARP–full form and p89 fragment of PARP; Tubulin–alpha-tubulin; p19/17 caspase-3–p19/17 fragments of caspase-3; h–hours, d-days. B–The histogram of flow cytometry (FC) analysis data for HEK293T cells treated with these nanoparticles at different concentrations and MG132 (2 μM). All experiments were performed at least three times. Results are presented as mean +/− standard deviation (SD). * *p* < 0.05: significant difference compared to control cells (Mann–Whitney U test).

**Figure 5 nanomaterials-11-02167-f005:**
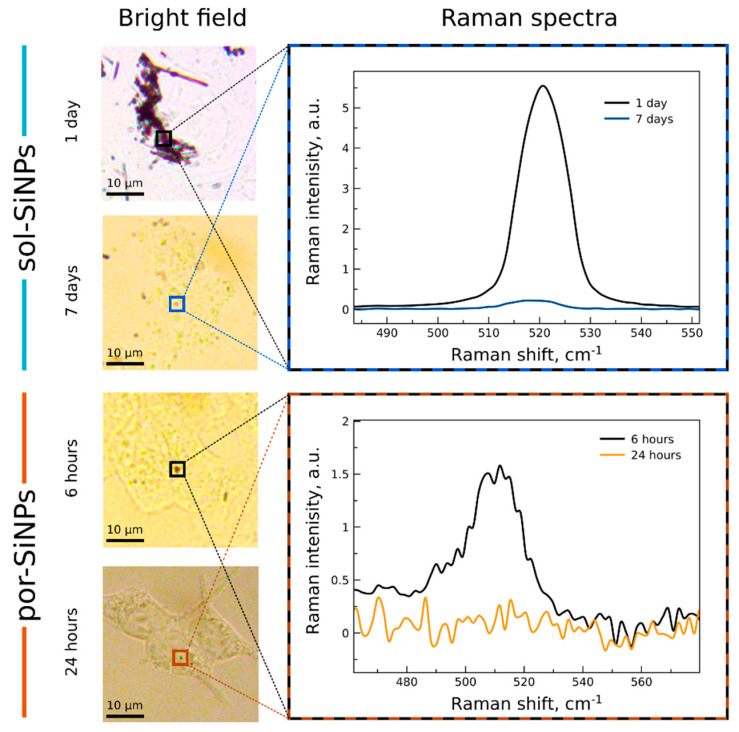
Widefield imaging and Raman spectra of SiNPs inside human cells (MCF7). **Left** side: widefield images of sol-SiNPs after 1 and 7 days of incubation and por-SiNPs after 6 and 24 h of incubation. **Right** side: corresponding Raman spectra of squared SiNPs.

**Figure 6 nanomaterials-11-02167-f006:**
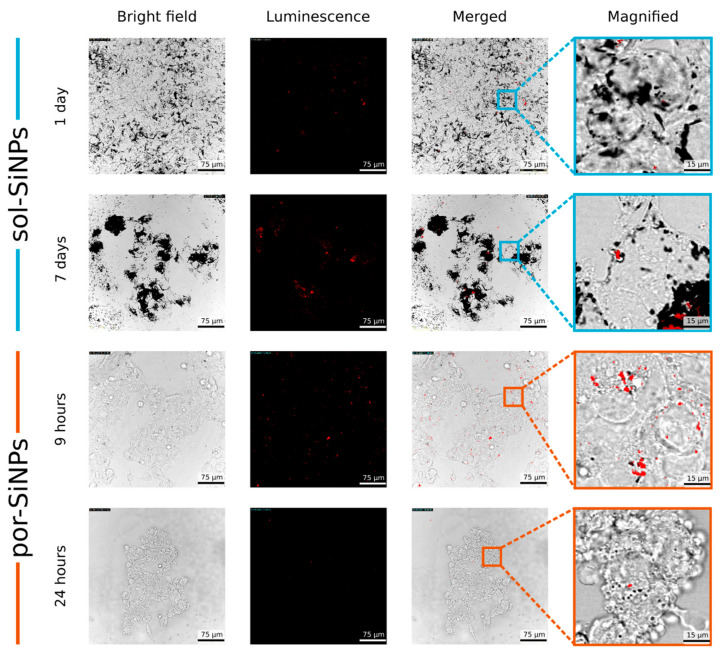
Widefield and luminescent confocal bioimaging of SiNPs inside human cells (HEK293T). Columns (from left to right): widefield view of cells; luminescence; merged images; magnification of the merged images. Rows: sol-SiNPs after 1 and 7 days of incubation, and por-SiNPs after 9 and 24 h of incubation.

**Table 1 nanomaterials-11-02167-t001:** Structural parameters of sol-SiNPs and por-SiNPs.

Sample Type	sol-SiNPs	por-SiNPs
Size of SiNPs (by DLS), nm	140	130
Size of nanocrystals (by TEM), nm	30	3
Zeta potential, mV	−50	−30
Specific surface area, m^2^/g	29	185
Pore volume, cm^3^/g	0.39	0.83
Pore diameter, nm	68	16

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
