# Peer review of "Optical Monitoring of the Biodegradation of Porous and Solid Silicon Nanoparticles"

_nanomaterials, 2021, doi:10.3390/nano11092167_

Round 1

Reviewer 1 Report

The authors described the synthesis of luminescent silica nanoparticles and investigate their optical properties, size and cell viability and cell imaging. This work is of some interest and it suitable for the scope of the Journal. However, the whole quality of the whole manuscript is still very poor. Before acceptance, some modifications should be made. The detailed comments are listed below.

  1. The size and size distribution of SiNPs should also determined by DLS.
  2. The fluorescent quantum yield and photostability should also be determined.
  3. The cell imaging of SiNPs should be improved.

Author Response

Reviewer #1:

The authors described the synthesis of luminescent silica nanoparticles and investigate their optical properties, size and cell viability and cell imaging. This work is of some interest and it suitable for the scope of the Journal. However, the whole quality of the whole manuscript is still very poor. Before acceptance, some modifications should be made. The detailed comments are listed below.

Our Response:

We are grateful to the Reviewer#1 for the detailed study of our manuscript.

Comment 1:

The size and size distribution of SiNPs should also determine by DLS.

Our Response 1:

DLS data were moved from Suppl. Info. to main part, i.e. into Figure 2C, 2F and also into Table 1 as summary.

All the changes are marked in yellow in the marked version of the manuscript (page 6, lines 246-259).

Comment 2:

The fluorescent quantum yield and photostability should also be determined.

Our Response 2:

Quantum yields of both sol-SiNPs and por-SiNPs photoluminescence were measured in previous studies and reach 0.2% and 2% for as-prepared samples, respectively. This was included into manuscript with the proper reference.

All the changes are marked in yellow in the marked version of the manuscript (page 8, lines 316-317).

Comment 3:

The cell imaging of SiNPs should be improved.

Our Response 3:

Cell imaging section was improved within two Figures S9 and S10 in Suppl. Info, which devoted to demonstration of internalization of SiNPs inside cells. That was proved by 1) 3D scan presented as 3 cross-sectional views, and 2) analysis of integrated PL intensity distribution within the depth (z-value). Both clearly show that SiNPs are localized in the center of the cell, not on the surface of Petri dish, or on top of the cell layer.

All the changes are marked in yellow in the marked version of the manuscript (pages 12-13, lines 446-458) and Suppl. Info. (Figures S9 and S10).

Reviewer 2 Report

Maxim Gongalsky and co-workers reported a manuscript that describes the optical property changes of porous and solid silicon nanoparticles during degradation. The authors mainly analyzed the optical properties of each nanoparticle, including photoluminescence (PL) and Raman spectra. However, there are some points that need to be addressed, so I want to recommend the major revision of this manuscript. My comments are described below. If possible, I would like to review this manuscript again after the revision.

1. There is an error in Figure 1. Where is the figure caption?
2. The preparation method of each particle (porous, solid) should be described in Figure 1. In this version, Figure 1 is difficult to follow the details.
3. I think the DLS data for each particle should be included in Figure 2. In addition, the detailed information about each particle should be included in the manuscript as a Table (DLS size, zeta-potential, nanocrystal size distribution, BET results, etc.).
4. I think the Raman spectra are not clear to show the crystal changes of nanoparticles. If possible, the authors should show the changes using XRD.
5. How about the cell viability upon treatment of Si(OH)4 in the given condition?
6. It is difficult to believe the Raman spectra and luminescent confocal imaging of SiNPs inside cells (Figure 5, Figure 6). How did the authors confirm that the particles are in the cells? It looks just aggregated on the surface of cells or stuck between cell culture dish and cells. Furthermore, the luminescent confocal image date looks very tricky. The author should show more data to prove their results are verified. Without addressing this comment, I cannot recommend the acceptance of this manuscript. 
7. How about the FTIR spectra of each particle? I think the degradation property and optical property of silicon particles have a close correlation with surface functionality. In addition, the author should mention the surface character of SiNPs in the introduction part as well as the results part.

Author Response

Reviewer #2:

Maxim Gongalsky and co-workers reported a manuscript that describes the optical property changes of porous and solid silicon nanoparticles during degradation. The authors mainly analyzed the optical properties of each nanoparticle, including photoluminescence (PL) and Raman spectra. However, there are some points that need to be addressed, so I want to recommend the major revision of this manuscript. My comments are described below. If possible, I would like to review this manuscript again after the revision.

Our Response:

We are grateful to the Reviewer#2 for the detailed study of our manuscript.

Comment 1:

There is an error in Figure 1. Where is the figure caption?

Our Response 1:

Caption of Figure 1 was probably missing due to formatting error. This flaw has been eliminated.

Comment 2:

The preparation method of each particle (porous, solid) should be described in Figure 1. In this version, Figure 1 is difficult to follow the details.

Our Response 2:

We agree that single scheme for both types of articles may look confusing, therefore Figure 1 was modified to show both sol-SiNPs and por-SiNPs formation.

Consequent changes are colored yellow in the marked version of the manuscript (page2, lines 84-85; page 3, lines 90-98).

Comment 3:

I think the DLS data for each particle should be included in Figure 2. In addition, the detailed information about each particle should be included in the manuscript as a Table (DLS size, zeta-potential, nanocrystal size distribution, BET results, etc.).

Our Response 3:

We agree that DLS data may be interesting in the main part of the article, so it was moved from Suppl. Info. to Figure 2C, 2F. Figure caption and description were also modified.

Some information such as size of SiNPs, size of nanocrystals and zeta-potential was added to Table 1.

All the changes are marked in yellow in the marked version of the manuscript (page 6, lines 254-259; page 7, lines 278-281).

Comment 4:

I think the Raman spectra are not clear to show the crystal changes of nanoparticles. If possible, the authors should show the changes using XRD.

Our Response 4:

It is well known that the low frequency shift and broadening in the Raman signal are explained by the phonon confinement model, related to the change of spectral efficiency of the Raman scattering on optical phonons in nc-Si. So, the Raman peak intensity is proportional to the concentration of silicon, whereas the Raman peak position is governed by the diameter of the crystalline silicon core

Due to big size of Si nanocrystals in sol-SiNPs (> 30 nm), there is no significant Raman shift for sol-SiNPs. Also there is no any Debye-Sherrer broadening (of XRD peaks) for them, so it cannot be used for monitoring of the biodegradation too. Therefore, we used only intensity of Raman spectrum.

For porous nanoparticles with tiny nanocrystals, an assessment of their biodegradation was added from the change in Raman spectra. For porous nanoparticles, the Raman spectra were re-measured.

All the changes are marked in yellow in the marked version of the manuscript (page 8-9, lines 340-254).

Comment 5:

How about the cell viability upon treatment of Si(OH)4 in the given condition.

Our Response 5:

Though sometimes excessive silicic acid promotes some cytotoxicity as a product of SiNPs biodegradation, in present study the toxicity of SiNPs was negligible within the used concentrations. Therefore, the question of SiOH4 citotoxicity is out of the scope of this article, however, we agree, that can be interesting for more detailed toxicological study of SiNPs.

Comment 6:

It is difficult to believe the Raman spectra and luminescent confocal imaging of SiNPs inside cells (Figure 5, Figure 6). How did the authors confirm that the particles are in the cells? It looks just aggregated on the surface of cells or stuck between cell culture dish and cells. Furthermore, the luminescent confocal image date looks very tricky. The author should show more data to prove their results are verified. Without addressing this comment, I cannot recommend the acceptance of this manuscript.

Our Response 6:

We agree that the presented results do not clearly demonstrate efficient internalization of nanoparticles inside cells, therefore we provided additional evidences for that, shown in Figures S9 and S10 in Suppl. Info.

First demonstration of SiNPs internalization in cells in shown in Figure S9. Confocal images of both sol-SiNPs and por-SiNPs show that they are located inside cells. This could be seen in 3D scans, which are presented as 3 orthogonal cross-sections demonstrating that SiNPs are neither deposited on Petri dish bottom, nor sedimented on the surface of cells.

Second demonstration of SiNPs internalization is based on integrated luminescence depth profile. Figure S10 shows that maximal PL intensity is reached approximately at the depth, corresponding to the average center of cells. If SiNPs are deposited on the Petri dish bottom or on top of cells, maximum of PL will be at highest or lowest «z», but not in between. The description of this point was added to discussion on p. 12 (colored yellow).

All the changes are marked in yellow in the marked version of the manuscript (pages 12-13, lines 446-458) and Suppl. Info. (Figures S9 and S10).

Comment 7:

How about the FTIR spectra of each particle? I think the degradation property and optical property of silicon particles have a close correlation with surface functionality. In addition, the author should mention the surface character of SiNPs in the introduction part as well as the results part.

Our Response 6:

We agree, that surface composition is important for nanotechnology. Here we provided additional FTIR spectra (Figure S3 in Suppl. Info), which show that both types of nanoparticles have relatively thick oxide layer, which can be explained by dissolution of Ag nanoparticles just after etching by NHO3 solution, which also creates stable Si oxide shell as a side effect. This point was added to discussion on p. 6 lines 271-276 (colored yellow).

Round 2

Reviewer 1 Report

The authors have addressed my comments. I recommend the acceptance in current time.

Reviewer 2 Report

It is acceptable for the publication.